# Prospects for the Development of Pink1 and Parkin Activators for the Treatment of Parkinson’s Disease

**DOI:** 10.3390/pharmaceutics14112514

**Published:** 2022-11-19

**Authors:** Alexander V. Blagov, Andrey G. Goncharov, Olga O. Babich, Viktoriya V. Larina, Alexander N. Orekhov, Alexandra A. Melnichenko

**Affiliations:** 1Laboratory of Angiopathology, Institute of General Pathology and Pathophysiology, 8 Baltiiskaya Street, 125315 Moscow, Russia; 2Center for Immunology and Cellular Biotechnology, Immanuel Kant Baltic Federal University, 6 Gaidara Street, 236001 Kaliningrad, Russia; 3Scientific and Educational Center for Industrial Biotechnology, Immanuel Kant Baltic Federal University, 2 Universitetskaya Street, 236040 Kaliningrad, Russia

**Keywords:** mitophagy, Parkinson’s disease, mitochondria

## Abstract

Impaired mitophagy is one of the hallmarks of the pathogenesis of Parkinson’s disease, which highlights the importance of the proper functioning of mitochondria, as well as the processes of mitochondrial dynamics for the functioning of dopaminergic neurons. At the same time, the main factors leading to disruption of mitophagy in Parkinson’s disease are mutations in the Pink1 and Parkin enzymes. Based on the characterized mutant forms, the marked cellular localization, and the level of expression in neurons, these proteins can be considered promising targets for the development of drugs for Parkinson’s therapy. This review will consider such class of drug compounds as mitophagy activators and these drugs in the treatment of Parkinson’s disease.

## 1. Introduction

Parkinson’s disease (PD) is a progressive neurodegenerative disease that predominantly affects dopaminergic neurons (which produce dopamine) in the substantia nigra. [1]. PD is the second most common neurodegenerative disease after Alzheimer’s disease (AD), with an incidence of approximately 0.5–1% among persons aged 65–69 years and up to 1–3% among persons aged 80 years and older [2]. It is assumed that with an increase in the general age of the population, the prevalence of PD will increase by more than 30% by 2030, which will lead to various costs for the global medicine and economy [3].

The pathogenesis of PD is primarily characterized by a loss of nigrostriatal dopaminergic innervation, although the resulting neurodegeneration is not limited to substantia nigra dopaminergic neurons alone, but can also affect other neurons located in different areas of the central nervous system [2]. One of the important problems in the fight against the spread of PD is the difficulty of diagnosing this disease. Currently, there are no effective diagnostic tests.Thus, it is not possible to diagnose the development of PD before the onset of clinical symptoms, which also complicates the treatment of the disease: at the time of diagnosis, 60 to 80% of dopaminergic neurons are already affected [4].

The complexity of PD is additionally due to the multifactorial and not completely clear etiology of this disease, which includes both genetic and environmental factors. It has been reliably noted that the average age of onset of the disease is 60 years, and the prevalence of PD also increases with age [5]. Currently, more than 20 genes have been discovered, mutations in which are potentially associated with the development of PD and are transmitted both by autosomal dominant inheritance (such genes as: SCNA, LRRK2 and VPS32), as well as by autosomal recessive inheritance (such genes as: PRKN, PINK1 and DJ-1) [2]. At the same time, there are studies showing the possible role of non-genetic factors in the development of PD. Thus, it has been proven that the prototoxin 1-methyl-4-phenyl-1,2,3,6-tetrahydropyridine (MPTP), which is used in the production of herbicides, causes PD in humans and monkeys [6]. Studies [7,8,9] have shown an inverse relationship between smoking and the risk of developing PD. The exact reason for the relationship between smoking and PD is not completely clear, but one hypothesis is the observation that activation of nicotinic acetylcholine receptors on dopaminergic neurons by nicotine or selective agonists has a neuroprotective effect, which has been demonstrated in experimental models of PD. Another study showed an inverse relationship between the level of caffeine consumption and the development of PD [10].

The main drug in the treatment of PD is Levodopa, which is an isomer of the amino acid from which dopamine is synthesized. It is able to penetrate the blood–brain barrier, thus restoring the lack of dopamine in patients with PD [11]. To increase the effectiveness of Levodopa, it is often taken together with carbidopa, a drug that prevents exterior metabolism of Levodopa, that is, protects it from being converted into dopamine outside the brain [11]. Additional drugs used in PD are dopamine agonists, which do not directly convert to dopamine, but mimic its effects (pramipexole, ropinirole) [11]. The third group of drugs are inhibitors of enzymes involved in the dopamine cleavage reaction (monoamine oxidase B and catechol-O-methyltransferase), which include selegiline, rasagiline, safinamide, entacapone and opicapone [12]. Despite the fact that these drugs help to alleviate the condition of patients with PD, they are not able to stop the progression of the disease. The search for such compounds is a promising direction in the fight against PD. In this review, we consider the potential therapeutic efficacy of Pink1 and Parkin protein activators (one of the key targets in the pathogenesis of PD) in the treatment of PD.

## 2. General Mechanism of Mitophagy

Mitophagy is a process of selective destruction of damaged and dysfunctional mitochondria [13]. Mitophagy is the complex process, which is characterized by the presence of successive stages: initiation of mitophagy with labeling of protein targets selected for destruction on mitochondria, absorption of mitochondria through fusion with autophagosome, and final sequestration in lysosomes occurring by hydrolytic degradation of mitochondria [13]. On the basis of the proteins involved in the process of mitophagy, mitophagy is subdivided into PINK1/Parkin dependent mitophagy (which will be discussed below) and receptor-mediated mitophagy, which is activated to external stimuli through proteins of the outer mitochondrial membrane, which in this case are mitophagy induction receptors: BNIP3L (BCL2 interacting protein 3 like), FUNDC1 (FUN14 Domain Containing 1) and others. Both types of mitophagy are also initiated in response to different stimuli [14].

PINK1 and Parkin are the best-known proteins that directly regulate the quality of working mitochondria. PINK1 is a serine/threonine kinase, and Parkin is a cytosolic ubiquitin E3 ligase [15]. In healthy mitochondria, which are not subject to destruction, the N-terminus of the PINK1 protein molecule moves to the inner mitochondrial membrane, while interacting with the TOM/TIM protein complex. The efficiency of transport of the N-site of the PINK1 molecule is related to the value of the mitochondrial membrane potential; at the same time, the C-terminus of the PINK1 molecule, which contains the kinase domain, is directed towards the cytoplasm. On the inner membrane of healthy mitochondria, PINK1 is partially degraded by mitochondrial processing peptidase (MPP) and presenilin-associated rhomboid protease (PARL). The initially intact PINK1 domain with kinase activity undergoes proteasomal degradation already in the cytosol [16]. Stress factors, including: depolarization of the mitochondrial membrane, dysfunctional state of the ETC (electron transport chain) complex proteins located in the mitochondrial matrix, increased mutagenicity of mitochondrial proteins inhibit the degradation of PINK1 and lead to the accumulation of undamaged molecules of this protein on the outer mitochondrial membrane due to disruption of the process of intermembrane transfer of the N-terminal domain of PINK1 to the outer mitochondrial membrane [17]. PINK1 molecules located on the outer mitochondrial membrane undergo homodimerization, which leads to autophosphorylation and, as a result, promotes the activation of kinase activity and improves binding to key PINK1 substrates: Parkin and ubiquitin [17]. Due to properties such as a rapid increase in the number of molecules on the outer mitochondrial membrane and the ability to be activated in response to mitochondrial stress, PINK1 is an effective sensor of mitochondrial damage and mitochondrial dysfunction.

Parkin, as mentioned above, is the ubiquitin E3 ligase and contains the ubiquitin like domain and four RING domains, which, due to intramolecular interactions, block the active site and exhibit competitive activity for binding to E2 ligase [15]. In mitochondrial damage or mitochondrial dysfunction, PINK1 activates Parkin using two different but similar variants: in the first case, activation occurs through binding to ubiquitin and its subsequent phosphorylation at the Ser65 position, which also interacts with Parkin and activates it; the second variant is based on the direct interaction of these mitophagic enzymes—PINK1 directly phosphorylates Parkin at position Ser65 in the Parkin ubiquitin-like domain, which leads to conformational changes in the Parkin protein and allows it to interact with E2 ligase, which in turn leads to triggering the ubiquitination reaction [17]. Parkin thus functions as an enhancer of the mitochondrial damage signal from PINK1, which increases the number of ubiquitin protein molecules on the mitochondrial membrane, which results in the recruitment of even more Parkin molecules to the mitochondria. Being recruited into mitochondria, Parkin labels various mitochondrial proteins with ubiquitin, which are located in different parts of the mitochondria [18]. The emergence of a large number of ubiquitin chains serves as a signal to attract and bind to the mitochondrial surface of autophagy mediators such as: OPTN (Optineurin), NDP52 (nuclear dot protein 52 kDa), RABGEF1 (RAB Guanine Nucleotide Exchange Factor 1), RAB7A (Ras-related protein Rab-7a) and RAB5 (Ras-related protein Rab-5A), which are also called adapter proteins. Light chain 3 (LC3) of microtubule-associated protein 1 recognizes and interacts with the adapter proteins indicated above, which subsequently leads to the formation of mitophagosomes, in which dysfunctional mitochondria undergo final degradation after the fusion of mitophagosomes with lysosomes [19]. There is evidence, that for PINK1 and Parkin, there are a number of naturally occurring activator proteins in the cell. These proteins, as well as their synthetic derivatives, can be considered as possible drug options for the treatment of Parkinson’s disease, which will be discussed in the following sections [20].

## 3. The Role of Mitophagy Disorders in the Development of Parkinson’s Disease

The fundamental function of mitochondria is the production of ATP molecules. Additional functions of mitochondria are: Participation in the metabolism of fatty acids and amino acids, the formation of cofactors and coenzymes, including NADH and FADH2, regulation of Ca 2+ homeostasis and initiation of internal apoptosis [21,22]. Mitochondria are not static organelles; they are subject to mitochondrial dynamics, which includes fusion, fission, intracellular transport, and mitophagy. These processes require the coordinated work of a large number of special proteins, mutations in which can disrupt the functioning of mitochondrial dynamics and, as a result, lead the development of various types of diseases, including PD [22].

To maintain the vital activity of neurons, the effective work of mitochondria is especially important. Neurons have a heterogeneous polarized structure, the main elements of which are the body of the neuron and the axon and dendrites, through which nerve impulses enter and are transmitted. For such structure, it creates different needs for different parts of the neuron in the supply of ATP energy, while the greatest energy costs occur in the places where the nerve impulse is transmitted: presynaptic and postsynaptic terminals, which requires the localization of a large number of functional mitochondria in these compartments [23]. The energy of ATP is used to perform important processes occurring in neurons: the mobilization of synaptic vesicles, the formation of an actin cytoskeleton for efficient transport of neurotransmitters and organelles within a neuron, the creation of an electrochemical potential for synaptic transmission of a nerve impulse, the capture, and recycling of neurotransmitters, and in addition to the regulation of Ca 2+ dynamics [23,24].

The need for a different concentration of working mitochondria in different compartments of the neuron, depending on their energy and metabolic needs, requires the correct and coordinated work of the processes of mitochondrial dynamics. Mitochondrial transport is required for the delivery of mitochondria to distant regions of the neuron and back to the soma; it is subdivided into retrograde and anterograde transport. The fusion of mitochondria ensures the creation of new healthy and larger mitochondria, due to the fact that when functional and dysfunctional mitochondria are combined, healthy mitochondria are formed [25], the negative side of fusion is the fact that the total number of mitochondria in the cell decreases. Mitochondrial fission most often precedes mitophagy, provided that damage is unevenly distributed as a result of fission, healthy mitochondria are also formed in addition to dysfunctional mitochondria [25]. Mitophagy, which occurs predominantly in the body of the neuron, leads to the destruction of “expired” mitochondria, thus increasing the proportion of functional mitochondria in the cell and preventing the risks for the cell associated with an increased concentration of dysfunctional mitochondria, including the initiation of apoptosis and the development of an inflammatory response.

In the pathogenesis of PD, the occurrence of mitochondrial dysfunction has been proven, especially associated with a defect in complex I of the mitochondrial respiratory chain, which produces up to 40% of the proton gradient for ATP synthesis, and is also the main source of ROS (reactive oxygen species), which is formed as a by-product of electron transfer reactions in respiratory chain [26]. The role of mutations in the mitophagy proteins Pink1 and Parkin has also been proven in the development of PD, which leads to disruption of mitophagy and, as a result, to the accumulation of dysfunctional mitochondria in the neuron body [27]. What does this mean for dopaminergic neurons? Mitochondrial dysfunctions, concentrated in one mitochondrion, accumulate in many mitochondria, which leads to the generation of a pathological condition. Firstly, the energy and metabolic balance inside the cell is disrupted due to disruption of the process of oxidative phosphorylation and, as a result, a decrease in ATP production, which over time leads to energy depletion of the neuron. Secondly, mitochondrial dysfunction causes an increase in the production of ROS, which are stress signaling molecules that cause damage to biological macromolecules (which further enhances mitochondrial dysfunction in case of damage), the development of an inflammatory response, which is noted during neurodegeneration, and the initiation of apoptosis, which leads to death of dopaminergic neurons [28,29]. In addition, the direct participation of the Pink1 protein in the inhibition of the progression of PD was noted through interaction with α-synuclein, the accumulation of which in neurons leads to the development of neurotoxicity and the initiation of apoptosis. The interaction between PINK1 and α-synuclein reduced the accumulation of α-synuclein in neurons. At the same time, as a result of the introduction of the G309D mutation into the gene encoding PINK1, this interaction was terminated [30]. A relationship was also shown between the mitophagy regulator Pink1 and the regulator of mitochondrial division DRP1 [31]. Pink1 initiated phosphorylation of DRP1 in S616, which led to the activation of mitochondrial division. In samples of patients with PD with mutations in Pink1, a decrease in DRP1 phosphorylation was noted, which led to an elongation of mitochondria in neurons and a decrease in their number. This, in turn, can reduce the transport capacity of mitochondria to move to the required cellular compartments and lead to a lack of mitochondria, energy deficiency and subsequent initiation of apoptosis. The general scheme of mitophagy involvement in the pathogenesis of PD is shown in Figure 1.

## 4. Features of Pink1 and Parkin as Targets for the Therapy of Parkinson’s Disease

### 4.1. Characterized Mutant Forms of Mitophagy Proteins

The presence of characterized mutations with an understanding of the protein dysfunctions they cause is an important criterion for the preliminary evaluation of molecular targets for which drugs will be developed. There are several key mutations in the PINK1 and Parkin proteins that lead to the development of PD. Thus, it was shown that the PINK1-I368N mutant could not bind to the outer mitochondrial membrane due to conformational changes in its enzymatic center, which led to blocking of the very first stage of mitophagy [32]. Homozygous nonsense mutation of PINK1 p.Q456X leads to premature stop codon and decreased levels of PINK1 mRNA, resulting in the formation of completely dysfunctional PINK1 proteins [33]. The p.G411S mutant variant forms dimers and can even localize on the outer mitochondrial membrane; however, a partial loss of kinase activity does not allow it to perform its function completely [34]. PINK1 phosphorylates Parkin in the region of the ubiquitin-like (UBL) domain, which changes its conformation to open and active [35]. It was found that mutations in the UBL domain: G12R, R33Q, and R42P led to a decrease in Parkin phosphorylation, which, in turn, led to impaired Parkin activation [36]. Two other substitutions in the UBL domain, G12R and T55I, lead to auto ubiquitination of Parkin, resulting in its degradation [36]. It should be noted that the dysfunctions of PINK1 and Parkin in PD are related not only to fixed mutations. For example, the disruption of mitophagy can be manifested by the accumulation of S-nitrosylated PINK1 (SNO-PINK1), which is an incorrect post-translational modification leading to inhibition of the kinase activity of this enzyme. It was found that increased production of SNO-PINK1 led to neuronal death [37]. Mutant forms and post-translational modifications of Pink1 and Parkin and their pathological mechanisms are summarized in Table 1.

### 4.2. Level of Expression of Mitophagy Proteins in Neurons

Better targeting of the selected protein target requires high expression of this protein in the pathological group of cells, which will provide a high concentration of the protein in the cell and a greater likelihood of drug binding. In a mammalian cell culture study, dopaminergic neurons were found to overexpress PINK1 and Parkin proteins [38]. However, this study was conducted on healthy neurons that did not show signs of degeneration. A number of studies using cell culture models suggest that PINK1 may play a neuroprotective role in some forms of stress, because overexpression of wild-type PINK1, but not mutant PINK1, protects against cell death caused by chemical stressors such as neurotoxin [39,40,41]. Most likely, in response to increased oxidative stress and depolarization of mitochondrial membranes of dysfunctional mitochondria, an additional increase in the expression of mitophagy proteins occurs, however, since mutant proteins cannot perform their proper functions, they accumulate as “dead weight” and can serve as a good target, except for those forms of mutant proteins, that are rapidly degraded.

### 4.3. Labeled Cellular Localization of Mitophagy Proteins

Knowledge of the predominant intracellular localization of therapeutic targets is an important step in drug development. The localization of PINK1 and Parkin differs: PINK1 is a mitochondria-targeted protein, while Parkin is predominantly contained in the nucleus, and can also be located in the cytoplasm and recruited into mitochondria immediately after the initiation of mitophagy from the PINK1 signal [42,43,44]. Therefore, therapeutic compounds directed to PINK1 and Parkin should have the same preferential localization as their targets. However, the submitochondrial localization of PINK1 should be taken into account depending on the level of mitophagy in cells [45]. Thus, in normally functioning dopaminergic neurons, where high energy costs require rapid renewal of mitochondria, PINK1 is localized in a high concentration on the outer mitochondrial membrane, due to the blocking of its import into mitochondria, where it serves as a signal for the activation of mitophagy reactions. In pathogenic dopaminergic neurons, PINK1 can accumulate in the mitochondrial matrix in the case of mutations at the N-terminus of PINK1; however, if mutations do not affect the ability of PINK1 to bind to the outer mitochondrial membrane, then it is also predominantly localized on it, if we additionally take into account the high needs for mitophagy, which not happening. For the Parkin protein mutants associated with the development of PD: ParkinR42P and ParkinG430D, inhibition of import into the nucleus was shown, which indicated the predominant cytoplasmic localization of these mutant forms, which, however, did not mean the abolition of nuclear localization for other Parkin mutants [46]. Additionally, taking into account the increased need for mitophagy in pathological neurons in PD, Parkin concentration on the outer mitochondrial membrane is possible if PINK1 is not subject to dysfunctional mutation.

## 5. Current State of Development of Pink1 and Parkin Activators

Since the deficiency of Pink1 and Parkin function is one of the factors in the development of PD, the development of pharmaceutical compounds that can restore the normal level of mitophagy, which will potentially help stop neurodegenerative processes, is promising in the fight against PD. Despite the well-known role of dysfunction of Pink1 and Parkin proteins in the pathogenesis of PD, there are currently only a few preclinical studies of the effectiveness of compounds activating the action of these proteins, so it will be possible to state what effect this therapy will have in the treatment of patients with PD no sooner than in 10–15 years. An important issue before the study of drugs in clinical trials is the assessment of the toxicity of selected compounds and the absence of an excessive effect of enhancing mitophagy, which can also negatively affect the vital activity of cells.

Mitophagy activators can be conditionally divided into three groups: Pink1 activators, Parkin activators, and inhibitors of ubiquitin-specific protease (USP30), whose function is inverse to that of Parkin [47]. One promising option for Pink1 activators is the use of kinetin, which is a precursor of the energy substrate kinetin triphosphate (KTP), which is an analog of ATP. In the study in a human neuron model, Pink1 was shown to bind to KTP with higher catalytic efficiency than to ATP. The introduction of kinetin led to an increase in the activity of wild PINK1 and mutant PINK1 G309D, and also inhibited neuronal apoptosis [48]. Kinetin had previously been shown to be well tolerated in humans and was found to freely cross the blood–brain barrier in a mouse model [49]. However, in a later study in rodent models, kinetin did not protect against α-synuclein-induced neurodegeneration, suggesting the need for more comprehensive preclinical studies [50]. There is also information about the effectiveness of natural compounds isolated from plant tissues. One such compound is celastrol, which reduced MPP+-induced death of dopaminergic neurons, reduced mitochondrial membrane depolarization, and increased ATP production in a cellular model of PD. In a mouse model, celastrol administration had a restorative effect on the motor symptoms of PD, slowed down neurodegeneration in the substantia nigra, and enhanced mitophagy. It was shown that celastrol was able to enhance the expression of PINK1 and some other proteins inhibited in PD [51]. Similar effects were also shown by another natural compound, salidroside [52]. Currently, there are no carried out in vivo studies on the effectiveness of Parkin activators, however, there are patented compounds that have shown their effectiveness in activating Parkin in vitro, which provides prerequisites for further development of this area of research [53]. Mediated activation of mitophagy through inhibition of USP30 may also be a promising option in the treatment of PD. USP30 is a convenient target located predominantly on the outer mitochondrial membrane. USP30 removes ubiquitin residues from labeled mitochondria, thus preventing mitophagy [47,54]. To date, several highly selective USP30 inhibitors have been identified that had shown their effectiveness in cell cultures by increasing the levels of ubiquitination and mitophagy [47]. However, the question of the effectiveness of these developments in in vivo studies is still open. Therapeutic strategies to restore mitophagy in PD are presented in Table 2.

## 6. Evaluation of the Prospects for the Development of Pink1 and Parkin Activators for the Treatment of Parkinson’s Disease

The development of drugs based on Pink1 and Parkin activators is a new direction and is currently still in its infancy, but it has the potential for further development. This is especially true for the development of Pink1 activators, the effectiveness of which has already been tested in animal models. However, it should be taken into account that before using the developed drugs, patients with PD should undergo genetic screening for the detection of Pink1 and Parkin mutations, since it is for patients with these mutations that a therapeutic effect in the treatment of PD is potentially possible. A more detailed analysis of the advantages and disadvantages of the development of Pink1 and Parkin activators for the treatment of PD is presented in Table 3.

## 7. Discussion

Developing of small-molecule-based Pink1 and Parkin activators is the most practical option for a relatively rapid release of such drugs. Some studies have focused on a different approach, based on increasing the expression of Pink1 and Parkin by introducing viral vectors containing wild-type genes of these proteins [55]. This approach has indeed been shown to be effective in reducing neurodegeneration in animal models, but seems far more distant and less realistic for clinical application in the treatment of PD. A potentially effective option for the search for Pink1 and Parkin activators can be provided by the use of in silico modeling. This is facilitated by the detailed structure of the open and closed conformations of the Parkin molecule with the resolution of each atom [56]. In addition to drug screening, in silico methods can help identify new upstream regulators of mitochondrial dysfunction in PD, as was shown in the identification of ATF4, which is a regulator of transcriptional changes identified in Pink1 and Parkin mutants [57]. One of the potential therapeutic strategies for the treatment of PD may be an increase in the expression of Pink1 and Parkin proteins. A number of studies in animal models have shown that increasing the expression of these proteins through gene augmentation reduced neurodegeneration, improved locomotor activity, and increased survival [55]. An issue requiring attention is the selection of the correct system of markers for assessing mitophagy in vivo directly in neurons. In cell culture, methods for studying mitophagy are effectively applied, but studies on animal models often show unsatisfactory results. This is primarily due to the fact that in vivo detection requires greater sensitivity than in vitro detection. Thus, the study [58] compared two markers for the effectiveness of studying the process of mitophagy in vivo. In addition to Pink1 and Parkin, other therapeutic targets are also considered, mutations in which also led to the development of PD: alpha-synuclein, DJ-1, VPS35, LRRK2, etc. [26]. Many of these proteins are also directly or indirectly involved in mitophagy. The development of complex drugs aimed at normalizing the functioning of several key mutant proteins can help patients who have more than one mutation associated with the pathogenesis of PD. In addition, it is necessary to study the effect of Pink1 and Parkin activators on other features of the pathogenesis of neurons in PD in addition to protection against neurodegeneration, namely, the reduction of neuroinflammation and the normalization of energy status with the restoration of ATP production. In contrast to similar reviews, for example, here we conducted a more detailed analytical work, which included not just a description of Pink1 and Parkin and currently available therapeutic developments, but also an analysis of the consideration of Pink1 and Parkin as therapeutic targets based on their properties, as well as an analysis of the prospects for the development of mitophagy protein activators for the treatment of PD [59].

## 8. Conclusions

To effectively combat the progression of Parkinson’s disease requires the development of new drugs that can greatly slow down or stop neurodegeneration. One of the promising targets for new therapeutic agents are the main mitophagy enzymes: Pink1 and Parkin. Mutant forms of Pink1 and Parkin, as well as associated dysfunctional states of these proteins, are well characterized. To date, there are several promising compounds that are mitophagy activators and have been shown to protect dopaminergic neurons from degeneration in cell culture and animal models. However, the clinical use of these drugs will not be expected until 10–15 years if the studied compounds pass all stages of clinical trials. It will be possible to use mitophagy activators in the treatment of Parkinson’s disease in patients with mutations in the Pink1 and Parkin proteins.

## Figures and Tables

**Figure 1 pharmaceutics-14-02514-f001:**
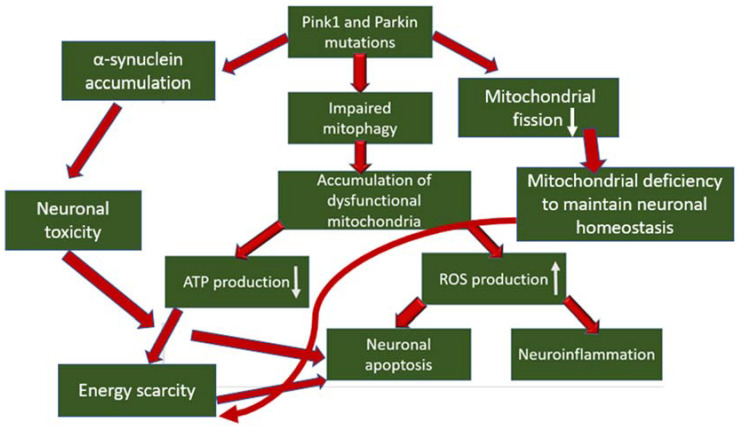
The general scheme of mitophagy involvement in the pathogenesis of PD.

**Table 1 pharmaceutics-14-02514-t001:** Mutant forms and post-translational modifications of Pink1 and Parkin and their pathological mechanisms.

Protein	Mutation or Modification	Pathological Mechanism
Pink1	I368N	Impossibility of binding to the outer mitochondrial membrane
Pink1	p.Q456X	Decreasing of mRNA level
Pink1	p.G411S	Partial loss of kinase activity
Parkin	UBL domain (1)	Decrease in Parkin phosphorylation
Parkin	UBL domain (2)	Parkin degradation
Pink1	S-nitrosylation	Loss of kinase activity
Pink1	G309D	Accumulation of α-synuclein

**Table 2 pharmaceutics-14-02514-t002:** Therapeutic strategies to restore mitophagy in PD.

Therapeutic Compound	Therapeutic Strategy	Therapeutic Mechanism
Kinetin	PINK1 activator	Direct binding to the active site
Celastrol	PINK1 activator	Enhancing expression
Salidroside	PINK1 activator	Enhancing expression
No name	Parkin activator	Direct binding to the active site
USP30i	USP30 inhibitor	Direct binding to the active site

**Table 3 pharmaceutics-14-02514-t003:** Main advantages and disadvantages of the development of Pink1 and Parkin activators for PD therapy.

Factor	Advantages	Disadvantages
Investment attractiveness		Approximately 5–10% of PD patients have monogenic forms of the disease. Mutations encoding genes in Pink1 and/or Parkin account for 1–9% of all genetic PD. Therefore, considering the low percentage of subjects bearing these mutations, it does not seem very attractive to invest money and time in the development of novel activators of mitophagy.
State of Development	Studies have shown the ability of potential drugs to reduce neuronal degeneration, which is a prerequisite for efficacy in the treatment of PD.	For Parkin activators and USP30 inhibitors, only results of single in vitro studies are available; for Pink1 activators, results on animal models are available. Due to the lack of ongoing clinical trials, the potential entry of drugs to the market will not occur earlier than in 10–15 years.
Choice of active compound	Small molecules that can be picked up relatively quickly by in silico methods. Small molecules are more likely to reach the target location of the mutant protein. High probability of detecting the active substance from natural compounds, which will simplify the production process.	High toxicity of synthetic compounds for humans is possible. An accurate determination of the effective dose of the active substance is required in order to avoid excessive activation of mitophagy.
Target selection	Mutations in Parkin and Pink1 are one of the main genetic factors in the development of PD. Most of the Parkin and Pink1 mutants are well characterized, which facilitates drug selection. Convenient localization of Pink1 in mitochondria.	PD therapy based on mitophagy enhancement is not suitable for all patients, since Parkin and Pink1 mutations are not the only factor in the development of PD. Some mutations in Parkin and Pink1 may make impossible recovery of protein function or can be reason for rapid degradation of the molecule. Not fully defined localization of mutant forms of Parkin. Mitophagy can be carried out without the participation of Pink1 and Parkin.

## Data Availability

Not applicable.

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
