# Peer review of "Prospects for the Development of Pink1 and Parkin Activators for the Treatment of Parkinson’s Disease"

_pharmaceutics, 2022, doi:10.3390/pharmaceutics14112514_

Round 1

Reviewer 1 Report

Mitochondrial dysfunction and defective mitophagy play a central role in the pathogenesis of PD. Pink1/Parkin are shown to mediate the degradation of damaged mitochondria via selective autophagy (mitophagy). However, Pink1/Parkin signaling is altered in PD. In this review article, the authors suggest that treatment with mitophagy activators may represent a promising therapeutic strategy for the treatment of PD patients carrying Pink1 and Parkin mutations, which are responsible for early-onset autosomal recessive forms of PD. Overall, the review provides useful information. Nevertheless, the authors should address several major compulsory issues.

The first half of the manuscript, especially from sections 1 to 4, can be significantly reduced since most of the information has already been reported or covered by many other articles. The paper needs to be more concise.

In addition, the review needs to include 1 or 2 elaborated figures, showing potential mechanisms of Pink1/Parkin-mediated neurotoxicity in PD.

The best-characterized ubiquitin-mediated mitophagy pathway involves PINK1 and Parkin. However, there are also different PINK1/Parkin-independent mitophagy pathways.

The authors need to discuss a key issue in a new section: why does Pink1/Parkin-induced mitophagy is easily inducible in cell cultures but is very challenging to observe in neurons?

The authors need to further discuss the therapeutic potential of Pink1 and Parkin overexpression in neurodegenerative diseases.

There is a lack of studies using compounds potentially effective upregulating the enzymatic activity of Pink1 and Parkin either in vivo models or (pre)clinical studies. Consequently, in silico modeling may represent a more suitable primary strategy for drug discovery. Some discussion should be devoted.

Approximately 5-10% of PD patients have monogenic forms of the disease. Mutations encoding genes in Pink1 and/or Parkin account for 1-9% of all genetic PD. Therefore, considering the low percentage of subjects bearing these mutations, it doesn’t seem very attractive to invest money and time in the development of novel activators of mitophagy.

Importantly, increased mitophagy is not directly related to an improvement of nigrostriatal dopaminergic degeneration in PD. So, this reviewer suggests removing the following statement “One of the promising targets for new therapeutic agents are the main mitophagy enzymes: Pink1 and Parkin, since the role of mitophagy disorders in the pathogenesis of Parkinson's disease has been proven.”

Spell out the names of the OPTN, NDP52, RABGEF1, etc. before using the abbreviations.

The authors need to considerably review the manuscript for sentence structure and grammatical errors.

Author Response

Response to Reviewer 1 Comments

Point 1: The first half of the manuscript, especially from sections 1 to 4, can be significantly reduced since most of the information has already been reported or covered by many other articles. The paper needs to be more concise.

Response 1: It was reduced.

Point 2: In addition, the review needs to include 1 or 2 elaborated figures, showing potential mechanisms of Pink1/Parkin-mediated neurotoxicity in PD.

Response 2: . We have expanded the general scheme in Figure 1.

Point 3: The best-characterized ubiquitin-mediated mitophagy pathway involves PINK1 and Parkin. However, there are also different PINK1/Parkin-independent mitophagy pathways.

Response 3: Added in more detail in section 2.

Point 4: The authors need to discuss a key issue in a new section: why does Pink1/Parkin-induced mitophagy is easily inducible in cell cultures but is very challenging to observe in neurons?

Response 4: Do you mean that it is difficult to detect mitophagy in neurons or difficult to induce? I wrote about the difficulty of in vivo detection (in Discussion – reference 59), correct me if you meant something else.

Point 5: The authors need to further discuss the therapeutic potential of Pink1 and Parkin overexpression in neurodegenerative diseases.

Response 5:  Information was added in Discussion (reference 58).

Point 6: There is a lack of studies using compounds potentially effective upregulating the enzymatic activity of Pink1 and Parkin either in vivo models or (pre)clinical studies. Consequently, in silico modeling may represent a more suitable primary strategy for drug discovery. Some discussion should be devoted.

Response 6:  Information was added in Discussion (reference 56,57).

Point 7: Approximately 5-10% of PD patients have monogenic forms of the disease. Mutations encoding genes in Pink1 and/or Parkin account for 1-9% of all genetic PD. Therefore, considering the low percentage of subjects bearing these mutations, it doesn’t seem very attractive to invest money and time in the development of novel activators of mitophagy.

Response 7: The statement was added in the table 3.

Point 8: Importantly, increased mitophagy is not directly related to an improvement of nigrostriatal dopaminergic degeneration in PD. So, this reviewer suggests removing the following statement “One of the promising targets for new therapeutic agents are the main mitophagy enzymes: Pink1 and Parkin, since the role of mitophagy disorders in the pathogenesis of Parkinson's disease has been proven.”

Response 8: .It was removed.

Point 9: Spell out the names of the OPTN, NDP52, RABGEF1, etc. before using the abbreviations.

Response 9: Abbreviations have been explained.

Point 10: The authors need to considerably review the manuscript for sentence structure and grammatical errors.

Response 10: Found errrors have been corrected.

Reviewer 2 Report

It's an interesting discussion on the prospect of mitophagy mediator Pink1/Parkin as the therapeutic target of Parkinson's disease, which could serve as a promising direction. The overall content is well-constructed, still, I have some minor suggestions that can be addressed in this manuscript:

1. The relationship between Pink1 and mitochondrial fission factor Drp1 and its adaptors in Parkinson's Disease, which had been identified in some papers, could be illustrated and integrated in this content.

2. The mutant form of Pink1/Parkin with their biological and clinical relevances and Pink1/Parkin mitophagy activators with their respective model of mechanisms and model systems mentioned in the content can be summarized in one or two concise Tables.

3. There are some published review articles with similar topics (e.g. PMID: 35311891), it would be great if major distinctions and novelties can be pointed out.

Author Response

Point 1: The relationship between Pink1 and mitochondrial fission factor Drp1 and its adaptors in Parkinson's Disease, which had been identified in some papers, could be illustrated and integrated in this content.

Response 1: The information was added in text (reference 31, line 221-228) and in figure 1 (expanded).

Point 2: The mutant form of Pink1/Parkin with their biological and clinical relevances and Pink1/Parkin mitophagy activators with their respective model of mechanisms and model systems mentioned in the content can be summarized in one or two concise Tables.

Response 2: Tables 1 and 2 were added.

Point 3: There are some published review articles with similar topics (e.g. PMID: 35311891), it would be great if major distinctions and novelties can be pointed out.

Response 3: The information was added in Discussion (reference 60, line 388-393).

Round 2

Reviewer 1 Report

The authors have addressed all my concerns.